# New Approaches to Old Techniques in Cell Handling for Microscopy

**DOI:** 10.3390/cells14161271

**Published:** 2025-08-18

**Authors:** Zhanna Bartosh, Veenu Aishwarya, Wayne W. Hancock, Tatiana Akimova

**Affiliations:** 1Division of Transplant Immunology, Department of Pathology and Laboratory Medicine, Children’s Hospital of Philadelphia, Perelman School of Medicine, University of Pennsylvania, Philadelphia, PA 19104, USA; bakss0312@gmail.com (Z.B.); whancock@pennmedicine.upenn.edu (W.W.H.); 2AUM Biotech, LLC, Philadelphia, PA 19104, USA; veenu.aishwarya@aumlifetech.com

**Keywords:** cytospin, microscopy, cell imaging, fluorescent microscopy, oligonucleotides, ASO

## Abstract

Appropriate concentrations of reagents, an absence of significant cell clumps and debris, minimization of artifacts and ensuring satisfactory cell preservation directly affect the quality of data generated and cannot be overestimated. Traditionally, cells in suspension are prepared using a cytospin, which uses centrifugal force to concentrate and deposit cells onto a glass slide. Adherent cells are traditionally grown on coverslips located on the bottom of the wells of cell culture plates, or using special chamber slide systems. In our laboratory, we developed and tested simplified homemade approaches for both techniques, allowing users to perform large volume cell functional tests followed by microscopy evaluation without a need for a cytospin, special chamber slide systems or the use of round cover slips. We present methods and illustrative examples involving the cellular uptake of self-delivering oligonucleotides in murine splenocytes and in two adherent human tumor cell lines.

## 1. Introduction

Cytospin technology helps to efficiently increase cell yield in samples with low cellularity [1,2]. However, this method requires careful tuning of centrifugal force and time to find a balance between reasonable cell adhesion to the slide and preservation of cell morphology [3,4]. Additionally, there are reports or poor performance of cytospin in comparison with different techniques, not involving centrifuge force [5,6,7]

In our early experiments, we found that primary lymphocytes, stimulated in vitro in functional tests, became too fragile to preserve their nuclei membranes fully intact in cytospin, while decreased force led to poor attachment and loss of cells. An additional barrier to preserving cellular morphology was the phagocytosis of microbeads and production of neutrophil extracellular traps (NETs) by human neutrophils. Distorted cell morphology, microbead detached from the neutrophil surface and inability to visualize NETs in cytospin slide preps encouraged us to reconsider our slide preparation techniques. We eventually found that any functional tests with primary cells in suspension, including human and murine lymphocytes and neutrophils, may be performed directly on charged microscopy slides [8].

A second challenge with primary cells in suspension was to prepare relatively large time-series of experiments to visualize cellular uptake and distribution of fluorescently labeled antisense oligonucleotides (ASO). Published protocols require use of formaldehyde fixed cells, while fluorescent conjugate of ASO and some antibodies targeting superficial cellular markers do not sustain fixation well. We combined flow cytometry staining, traditional cell culture in Falcon test tubes and slide preparation using unfixed cells. This combination allowed us to prepare time-series experiments without cytospin or time-lapse imaging systems.

Adherent cells are less demanding for microscopy than cells in suspension, and itcommonly involves culturing them on coverslips in the wells of cell culture plates or in chambered microscopic slides [9]. However, this method becomes time consuming and relatively expensive in extensive experiments when new compounds are tested in time-series. Our current microscope with cell imagining system requires the use of slides faced down, which excludes the imaging of multiwell microplates. Using a previously developed modification of cell culture directly on microscopy slides, we further elaborated it to create a multiwell home-made analog of chamber slides.

Our modifications are simple and straightforward, allowing researchers to save time, efforts and money during cell preparation for microscopy. An additional advantage of our technique is the use of standard laboratory equipment and a standard fluorescent microscope.

## 2. Materials and Methods

### 2.1. Primary Cells and Cell Lines

Murine splenocytes were isolated from 6- to 8-week-old C57BL/6 mice (The Jackson Laboratory, Bar Harbor, ME, USA). The A549 lung carcinoma cell line was obtained from NCI/NIH, and HEp-2 cells were obtained from Cytion (Eppelheim, Germany, cat # 300397).

### 2.2. Cell Culture

Single cell suspensions from spleens were prepared as described [10] and cryopreserved with Cryostor CS5 cell cryopreservation media (Biolife Solution, Bothell, WA, USA cat# 205102), or used immediately used for functional assays. Cell lines and cryopreserved splenocytes were thawed in RPMI1640 (Invitrogen, Waltham, MA, USA), supplemented with 10% heat-inactivated FBS. Cell lines were recovered after cryopreservation for 2 days in the flasks (T25, cat#130189, ThermoFisher Scientific, Waltham, MA, USA), counted and reseeded in the wells of microscopy slides (see below). Cell growth media for A549 cells: RPMI, supplemented with 10% FBS, Penicillin-Streptomycin (cat# 15140122, Gibco, Waltham, MA, USA) and Glutamax (cat# 35050061, Gibco). Cell growth media for HEp-2 cells: Advanced MEM (cat#12492013, Gibco), supplemented with 10% FBS, Penicillin-Streptomycin, Glutamax and 10 mM Hepes (cat# 15630130, Gibco). Later experiments show that the recovery step may be omitted, so cells may be seeded directly onto the slides in concentration 2000 per well in cell growth media. Splenocyte culture medium: RPMI, supplemented with 10% FBS, Penicillin-Streptomycin (cat# 15140122, Gibco) and 2-mercaptoethanol (100 µM, Gibco, cat#21985023).

For ASO uptake experiments, we used DMEM (cat# 10565042, Gibco), supplemented with 3% of FBS, with an addition of 3 mM NH4Cl and 1 μM Arsenic to enhance gymnosis, as reported [11]. Flow cytometry staining buffer: DPBS (cat# 14190144, Gibco) with 1% of FBS.

### 2.3. Preparation of Time-Series of Slides with Primary Cells in Suspension

We found that Superfrost Plus microscope slides (cat# 22-037-246 Fisher Scientific, Hampton, VA, USA) served well for our purposes, allowing reliable cell adhesion without a need for additional coating. Microscope slides were labeled, rinsed in DI water and dried lying flat in a Class II biological safety cabinet with its ultraviolet lamp on.

Splenocytes were incubated with FITC-conjugated anti-mouse CD45 antibody (cat# 103108, Biolegend, San Diego, CA, USA), 3–5 μL per 1 × 10^6^ cells, 4 °C, 30 min, in Falcon 5 mL round-bottom tubes, then washed in flow cytometry buffer. Further experiments showed that the same approach with “flow cytometry” staining may be used for other markers of interest, such as CD4, CD8, CD11b etc., providing there was relatively abundant surface expression on the cells.

Far-red fluorescent labeled Scramble ASO (AUMsilence, AUM Biotech, LLC, Philadelphia, PA, USA) was diluted in 500 μL of freshly prepared and pre-warmed (37 °C) “ASO uptake” DMEM media at a concentration of 1.5 μM was added to the splenocyte pellet. Resuspended cells were incubated in cell culture CO_2_ incubator in single tube, upright position, with a lid on. After 15, 30, 60, 90 and 120 min, cells were resuspended by shaking or inverting the tube, and aliquots of cell suspension were withdrawn. The volume of each aliquot depended on the number of slides to be prepared. We found that 10 μL of cell suspension at a concentration of 1–5 × 10^6^ cells/mL was sufficient for one slide, but when needed, cells may be concentrated by brief and gentle centrifugation (3 min, 300× *g*). When needed, aliquots of the same cells may be fixed (to prevent further biological activities) and stored in flow cytometry buffer for flow cytometry. For far-red labeled AUMsilence Scramble ASO, IC Fixation Buffer (cat# 00-8222-49, Invitrogen) should be used instead of formaldehyde-based fixation.

We added 10 μL of cell suspension as several spots in a row to the middle of a slide. These spots were then immediately smeared using the side of a 1 mL pipette tip. The slides were then placed on a hot plate (low heat, 55–60 °C for 20 min, with protection from light). We found that excessive liquid readily evaporates on the hot plate, eliminating the need for cell concentration. Skipping the centrifugation step also allows for a more precise time point snapshot and minimizes additional stress on the cells, leading to improved morphology. Furthermore, we observed that any residual protein and salt traces were negligible and dissolved easily during subsequent steps.

On the final stage, we used a hydrophobic barrier pen on the edges of the slides surrounding cells, and fixed cells with IC Fixation Buffer (cat# 00-8222-49, Invitrogen) at room T for 5 min, then gently removed fixation solution from the corner of each slide using a pipettor, and mounted slides along with nuclear staining, using Mounting Medium with DAPI (SlowFade Diamond Antifade Mountant, cat #S36968, Invitrogen).

Alternatively, primary cells may be incubated directly on Superfrost Plus microscope slides in cell culture media, as we reported [8]. Slides were prepared by a short rinsing in DI water followed by drying with ultraviolet lamp, as above, but hydrophobic barrier pen (GnomePen Classic Liquid Blocker, Newcomer Supply, Waunakee, WI, USA, cat#6507B) was applied around the edges prior to rinsing, to create a container for liquid. GnomePen has an advantage over analogs by being insensitive to detergents, which may be critical if a researcher uses a cell staining protocol requiring fixation and permeabilization steps.

Surprisingly, not only neutrophils, but also lymphocytes, incubated on Superfrost Plus slides for 30 min or longer, adhered strongly enough to remain in place when the culture media was gently removed by pipetting from the corner of slide. Following this step, the slides were heat-fixed and stained as needed.

### 2.4. Comparison of Air-Dried, Heat-Dried, and Immediately Fixed Slides

Murine splenocytes we incubated on Superfrost Plus microscope slides, prepared as described in Section 2.3, at a concentration of 1 × 10^6^ cells per slide for 30 min in splenocyte culture medium. After incubation, the medium was gently removed from the corner of each slide, and the cells were subjected to one of the following conditions:(a)Left to air-dry at room temperature for 10 min;(b)Heat-dried on a hot plate at low heat (55–60 °C) for 20 min;(c)Immediately fixed in fixation buffer (BD Pharmingen, San Diego, CA, USA, cat# 562574) for 50 min at 4 °C.

Slides in groups (a) and (b) were subsequently fixed using the same fixation buffer, and all downstream procedures were identical across groups. Following fixation, cells were permeabilized for 15 min at 4 °C (BD Pharmingen, cat# 562574), then blocked with UltraCruz Blocking Reagent (Santa Cruz, Dallas, TX, USA, cat# sc-516214) for 15 min. Slides were incubated overnight at 4 °C in a humid chamber with an anti-β-actin antibody (Sigma-Aldrich, St. Louis, MO, USA, cat# A5316). The following day, slides were washed with permeabilization buffer and incubated with Alexa Fluor^®^ 488-conjugated goat anti-mouse IgG antibody (BioLegend, cat# 405319) for 40 min at 4 °C. After washing, slides were incubated with propidium iodide (PI, 10 μg/mL in PBS) for 5 min, followed by two washes: first with permeabilization buffer and then with DPBS. Finally, slides were mounted using SlowFade Diamond Antifade Mountant (Invitrogen, cat# S36967).

### 2.5. Comparison of Cytospin Preparation and Cell Smears

Murine splenocytes were incubated in cell culture tubes containing splenocyte culture medium for 3 h, in the presence or absence of 3 ng/mL of phorbol 12-myristate 13-acetate (PMA) and 1 μM ionomycin. After incubation, 1 × 10^6^ cells were split into two equal portions (0.5 × 10^6^ cells each) for preparation via standard cytospin or cell smears, as described in Section 2.2.

Cytospins were prepared by centrifuging 0.25 × 10^6^ cells per well at 1000 RPM for 5 min in 100 μL of DPBS supplemented with 3% FBS. Smears were prepared as described in Section 2.3. Following preparation, one part of the slides was stained for histone H3 and CD45. Cells were fixed and permeabilized as described in Section 2.4, then blocked with UltraCruz^®^ Blocking Reagent. Slides were incubated overnight at 4 °C in a humid chamber with an anti-histone H3 antibody (Abcam, Cambridge, UK, cat# ab1791). The following day, slides were washed with permeabilization buffer and incubated for 40 min at 4 °C with Alexa Fluor^®^ 647-conjugated goat anti-rabbit IgG antibody (Invitrogen, cat# A-21245). After washing, slides were incubated for 30 min at 4 °C with FITC-conjugated anti-mouse CD45 antibody (BioLegend, cat# 103108), diluted 1:50 in DPBS containing 3% FBS. Slides were washed again in DPBS with 3% FBS and mounted using Mounting Medium.

The second parts of the slides were stained for β-actin and PI as described in Section 2.4.

### 2.6. Preparation of Time-Series of Slides with Adherent Cells

Using the previously established protocol for short time culture of primary cells directly on Superfrost Plus microscope slides (Section 2.3), we further developed this approach to create an analog of chamber slides system with multiple wells. To do so, we applied a single hole puncher to parafilm strips, pre-cut according to the size of microscopic slides, as shown at Figure 1A, then applied strips to the slides with light pressing (Figure 1B), removed the paper layer and set slides on the laboratory heater at medium power, 30 s to 1 min, until parafilm melted (Figure 1C). Slides were then rinsed slides in DI water and dried under an ultraviolet lamp, as above. A large lot of slides may be prepared in advance and then stored in closed plastic bags for future use; in that case, rinsing and ultraviolet treatment should be performed prior to cell culture. When needed, slides with parafilm wells may be washed by immersing them in 70% ethanol or 3% bleach solution, followed by rinsing with deionized water. These procedures do not affect the structural integrity of the wells or the adhesive properties of Superfrost Plus slides.

The relatively small volume of each well in this system is advantageous to save money on expensive reagents for the short-time culture experiment, but it also poses a risk of drying in case of overnight (or longer) protocols. To prevent that, slides were placed into standard 9 cm Petri dish plates, on top of the plastic “coasters”, made from the single-use soft disposable serological pipettes (Figure 1D). DPBS or any similar buffer, added to the bottom of Petri dishes, prevents loss of liquid in the wells with cells (Figure 1E). Alternatively, single-hole punchers with smaller or larger diameters can be used as needed, which will affect the volume of the corresponding wells. In our protocol, a 6 mm hole diameter yielded a working volume of approximately 30 μL.

After thawing, A549 or HEp-2 cells were labeled for 5 min with CFSE 5 μM CFSE, (Invitrogen, cat# C1157) as described [10], and seeded directly to the slides at a concentration of 2000–3000 cells per well. The volume of each well was 30 ul. Next day, we evaluated slides under a microscope to ensure good cell morphology, CFSE fluorescence, sufficient attachment and 70–80% confluency of monolayer, then replace cell growth media with freshly prepared, warm ASO uptake DMEM media with 1.5 μM of far-red Scramble ASO. For this experiment, we started with the wells for longest incubation, 180 min, then replaced a cell culture in the wells for 150 min, then started ASO incubation in wells for 120 min, and so on, so the cells in all wells on all slides were ready at the same time, 180 min after the start of incubation in the longest time wells. Then we removed ASO by rinsing slides shortly in room T DPBS, dried them on the hot plate (low heat, 55–60 °C, 20 min, protect slides from the light), and removed parafilm using forceps. Then we applied hydrophobic barrier GnomePen around wells and proceed with cell fixation and DAPI staining as above, protected from the light. Use of a hydrophobic barrier may be an unnecessary precaution, since slides were still hydrophobic on the areas covered with parafilm (Figure 1F).

Slides were mounted as above and evaluated with a fluorescent microscope Keyence BZ-X700 (Itasca, IL, USA) and images captured using BZ-X Analyzer (Itasca, USA).

### 2.7. Use of Generative Artificial Intelligence in Graphical Abstract

We used ChatGPT (GPT-5) to generate images of cells in suspension, a cytospin centrifuge, and a microscope. Gemini AI (2.5) was used to create images of adherent cells on a glass surface, coverslip cell cultures in a plate, and cell culture tubes. The Image_Creators (v1) tool on the Poe AI platform was used to generate an image of a chamber slide system.

### 2.8. Data Handling and Statistics

We evaluated cells in each experimental condition, 3–4 different fields of microscopic view, and assessed at least 100 cells in details. Resulting data with dynamics of ASO cellular uptake and distribution were calculated as % of the total, and plotted against time using GraphPad Prism 8. Each experiment was performed at least twice. To compare the frequency of ASO cytoplasmic granules in A549 vs. HEp-2 cells, we performed 7 two sided Fisher’s exact tests in 7 contingency tables with raw counts of corresponding cells, one table for each incubation time (i.e., 15, 30, 60, 90, 120, 150 and 180 min), and then we applied Bonferroni correction for all *p* values. *p* values of less than 0.05 (after correction) were considered significant.

## 3. Results

### 3.1. Comparison of Different Techniques for Slide Preparation

We compared cell yields and the preservation of cellular morphology in murine splenocytes incubated on slides for 30 min, followed by one of three processing methods: air-drying at room temperature, heat-drying at 50 °C, or immediate fixation using a buffer containing formaldehyde and methanol. Slides that were air- or heat-dried showed noticeably higher cell yields, likely due to improved cellular adhesion (Figure 2A). Slides with immediately fixed cells showed lower—though still sufficient—cell numbers (Figure 2A, right).

All three methods preserved cellular morphology comparably, as assessed by β-actin (cytoplasmic) and propidium iodide (PI; nuclear) staining. Notably, air-dried and heat-dried slides allowed for the detection of apoptotic cells, which were PI-positive but β-actin–negative (Figure 2B, arrowed). This may reflect the lack of active adhesion in apoptotic cells: they likely adhered passively during the drying process but were lost during washing when fixation was performed immediately.

In another set of experiments, we compared the performance of cytospin slides and cell smear slides. Splenocytes were cultured for 3 h in the presence or absence of PMA and ionomycin, then divided into equal portions and processed for microscopy using either the cytospin or smear technique.

Staining for β-actin and propidium iodide (PI) revealed that both methods preserved cellular morphology and yielded comparable cell numbers, although slight differences in staining patterns were observed (Figure 3). Non-stimulated splenocytes exhibited similar morphology and staining patterns in both cytospin and smear preparations. However, activated lymphocytes in smear slides showed a pronounced accumulation of β-actin staining at one pole of the cell (Figure 3, arrowed), potentially reflecting cytoskeletal reorganization in response to stimulation [12,13]. This polarized staining pattern was not observed in cytospin-prepared cells, suggesting that the centrifugal force used during cytospin may disrupt cytoskeletal structures or mask subtle morphological changes.

Additionally, staining for histone H3 and CD45 was inferior in cytospin preparations compared to smears. In particular, we observed staining artifacts in some stimulated splenocytes, including apparent “leakage” of the histone H3 signal beyond the cellular boundaries (Figure 4, arrowed). This observation is consistent with our earlier experiments referenced in the Introduction, where we noted that the membranes of some stimulated lymphocytes become fragile and more susceptible to mechanical stress, such as that induced by centrifugal force, leading to increased artifact formation.

In conclusion, air-drying or heat-drying steps appear to enhance cell yields and enable the detection of apoptotic or dead cells. In contrast, immediate fixation on microscope slides led to the loss of nearly all dead cells and a portion of viable cells. Furthermore, cytospin preparation of activated primary cells may introduce staining artifacts, likely due to the effects of mechanical stress.

### 3.2. ASO Uptake in Primary Cells

We evaluated cellular morphology to confirm that different types of murine splenocytes, particularly lymphocytes, were not lost during cytospin-free slide preparation. As shown in Figure 5A, the typical morphological features of murine monocytes/macrophages, lymphocytes, and neutrophils were clearly identifiable. As described in Section 2, cells can also be labeled with lineage specific flow cytometry-grade antibodies to identify different cellular subsets as CD4+, CD8+ or B cells.

In the next step, we identified distinct patterns of ASO distribution within murine splenocytes: no ASO uptake, a few discrete fluorescent ASO spots, diffuse ASO uptake with low or high intensity, diffuse ASO uptake accompanied by small high-intensity spots, and diffuse ASO with relatively large granules (Figure 5B).

We then assessed the dynamics of ASO uptake and intracellular distribution over time (Figure 6). Remarkably, within just 15 min, approximately 80% of splenocytes exhibited intracellular ASO fluorescence. The most common pattern observed between 15 and 180 min was a diffuse, low intensity ASO signal. This data suggests that AUMsilence ASO were not concentrated in lysosomes following cell uptake. An absence of nuclear exclusion at early time points, with diffuse ASO signal throughout the cells suggests that ASO uniformly enters the nuclei of primary cells. The proportion of cells lacking detectable ASO uptake was highest at 15 min, then gradually declined across all cell types, reaching a minimum of 3.5% at 120 min and then increased to 13% at 180 min (Figure 6B). Concurrently, we observed a progressive increase in the percentage of cells containing bright, small ASO-positive spots—from 2.5% at 30 min to 24% at 180 min. At 120 min and later, percent of cells with large granules increased to 8%, along with decreased intensity of diffuse ASO signal. Notably, cells with the morphology of neutrophils (Figure 5A, bottom row) showed the most pronounced reduction in diffuse signal intensity, accompanied by accumulation of ASO in relatively large granules and apparent nuclear exclusion, features that were less prominent in lymphocytes.

Overall, we illustrated here that primary cells in suspension can be visualized on microscopic slides with satisfactory quality, and cell preparation for microscopy does not require the use of cytospin. ASO uptake and distribution differs in lymphocytes, monocytes and neutrophils, but the vast majority of primary cells efficiently uptake and keep ASO in their cytoplasm and nuclei.

### 3.3. ASO Uptake in Cell Lines: A549 and HEp-2 Cells

In contrast to primary cells, both cell lines exhibited 100% ASO uptake as early as 15 min post-incubation, with sustained intracellular presence observed for at least 180 min. We identified multiple patterns of ASO distribution, including both low and high nuclear fluorescence, as well as variable cytoplasmic staining intensity (bright vs. dim). In addition, we observed distinct features such as large cytoplasmic granules, enhanced ASO staining at the cell periphery, and accumulation of large ASO granules within the cytoplasm (Figure 7).

Interestingly, the dynamics of ASO uptake and intracellular distribution differed between the two cell lines. A549 cells, a human lung adenocarcinoma cell line, exhibited a pronounced accumulation of small ASO-containing granules at the cytoplasmic periphery as early as 15 min post-incubation. Simultaneously, ASO fluorescence was already detectable in the nuclei of all cells, displaying either bright or dim nuclear staining. These findings suggest that in A549 cells, AUMsilence ASO was not accumulated in lysosomes and efficiently enters the nuclei. The peripheral granules observed may reflect ASO localization within transport vesicles or endosomal compartments during intracellular trafficking (Figure 8A,B). At 30 and 60 min, the majority of A549 cells exhibited a nearly uniform, bright nuclear ASO signal accompanied by a low-intensity cytoplasmic signal, with few or no visible ASO granules. Bright nuclear staining remained the predominant pattern of ASO distribution in A549 cells during the first 120 min. After 60 min, the proportion of cells displaying low-intensity nuclear ASO signal gradually increased, reaching its peak at 180 min. Occasionally, A549 cells showed partial, but not complete, nuclear exclusion of ASO, often accompanied by the presence of large ASO granules either surrounding the nucleus or near the cytoplasmic periphery (Figure 8C). The highest percentage of cells exhibiting nuclear exclusion was observed at 60 min (5.6%), which then declined to 3% at 150 min and was no longer detectable by 180 min.

HEp-2 cells were originally thought to be derived from an epidermoid carcinoma of the larynx, but later it was established that HEp-2 cells are HeLa cells (cervical cancer) derivative. In contrast to A549 cells, HEp-2 cells displayed few visible ASO granules in the cytoplasm at 15 min. Instead, all cells exhibited a diffuse, low-intensity ASO signal in both the nucleus and cytoplasm. This type of ASO distribution was the most common at all times up to 180 min. Bright nuclear ASO staining increased from 14% at 90 min to 39% at 150 min (Figure 9A,B). As we observed in A549 cells, HEp-2 cells demonstrated brighter ASO signal in the cytoplasm on the periphery within the first 30–60 min. In contrast to A549 cells, HEp-2 seemed to accumulate ASO in the cytoplasm as large granules, especially at 15–120 min (Figure 9C). At 60 min, a small proportion of HEp-2 cells (1.5%) exhibited partial nuclear exclusion of ASO. This percentage increased to 4% between 90 and 150 min, before declining to 1.8% at 180 min (Figure 9D, left, arrowed), following a similar trend to that observed in A549 cells but with a delayed peak. By 180 min, some HEp-2 cells accumulated small ASO granules adjacent to the nucleus, in a location consistent with the Golgi complex (Figure 6A, 180 min; Figure 9D, right, arrowed). Notably, the nuclear ASO signal in these cells did not appear to be apparently reduced.

Overall, we report here that adherent cells can be successfully cultured for several days in small wells of a custom-made chamber slide system, which helps conserve reagents, media, and other laboratory supplies. The resulting cell monolayers maintain good quality and preserve cellular morphology sufficiently to enable high-resolution visualization of ASO uptake and distribution over time. This includes tracking the accumulation of ASO in cytoplasmic granules, nuclear exclusion, and changes in signal intensity at the level of individual cells. Importantly, ASO uniformly reaches all adherent cells within the first 15 min, and stays in the cell nuclei and cytoplasm for at least 180 min. The two tested cell lines, A549 and HEp-2, exhibited distinct temporal dynamics and spatial patterns of ASO distribution, confirming the importance of testing and validating any new active compound with at least two different cell lines.

## 4. Discussion

Microscopy imaging techniques enable acquisition of valuable information concerning cellular processes, molecules and organelles. Novel computational techniques can further improve image quality, resolution, and information extraction. However, the very first and the most basic step of this process is to obtain microscopic slides with the cells of appropriate quality and quantity.

There are several established methods for attaching non-adherent cells to microscope slides. One of the simplest and most straightforward approaches is to enhance cell adhesion by culturing under serum-free conditions [14,15]. However, serum deprivation can negatively impact cell viability and induce various alterations in morphology, metabolism, and gene expression [16,17,18,19]. The most commonly used method is cytocentrifugation (cytospin), which offers clear advantages in concentrating cells and preserving morphology. Nevertheless, cytospin may be inappropriate in certain contexts. In this study, we demonstrate that centrifugal force can adversely affect cellular morphology and lead to staining artifacts, particularly in activated primary cells. These issues may be mitigated by fixing the cells prior to cytospin preparation. However, this workaround also presents challenges, as fixation methods can significantly influence both cellular morphology and biochemical properties [20,21].

In this study, we demonstrated that Superfrost Plus microscope slides provide reliable adhesion of primary cells without the need for additional surface coatings. As we illustrated, the examination of apoptotic and dead cells requires air-drying or heat-drying steps to prevent their loss during staining and washing procedures.

For adherent cells, various microscopy preparation techniques have been described, including cytospin and paraffin-embedded cell blocks [5]. Our method, using parafilm and a hole puncher to create a custom chamber system, is notably simpler than other reported approaches, such as silicone rings pressed against the slide surface [22], and is much less expensive than commercially available small-well chambers.

We demonstrated that both cytospin-free cell adhesion and the parafilm chamber system can be successfully applied in large time-series experiments, such as studying the cellular uptake of self-delivering AUMsilence ASO. AUMsilence ASOs are characterized by a phosphorothioate backbone and flanking nucleotides with a 2′-deoxy-2′-fluoro-beta-d-arabinose furanose modification. Although this chemical structure has previously been reported to support self-delivery into both cell lines and primary cells [23,24,25,26,27], the precise mechanisms underlying ASO uptake and intracellular distribution remain poorly understood [28,29,30]. In this study, we showed that over 80% of primary splenocytes and nearly 100% of A549 and HEp-2 cells exhibited efficient ASO uptake. However, the temporal dynamics and intracellular distribution patterns varied between individual cells of the same type and across different cell types. These differences may be influenced by variability in plasma membrane proteins [31], endosomal proteins [32], chaperone proteins [33], receptors [34] and other yet unidentified factors. The microscopy-based approaches presented here may serve as valuable tools for future studies aimed at elucidating these mechanisms.

## 5. Conclusions

In the current manuscript, we presented modifications of methods of cell preparation for microscopy that allow researchers to simplify the process, making it more affordable and scalable. We illustrate the use of our methods by detailed evaluation of time-series experiments with cellular uptake of self-delivering AUMsilence ASO and outlined perspectives and modifications of our techniques to study certain types of cells or different reagents in variety of cellular processes.

## Figures and Tables

**Figure 1 cells-14-01271-f001:**
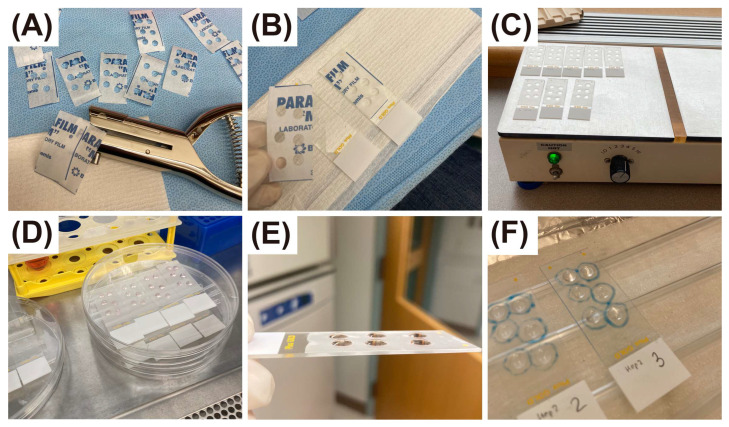
Chamber slides system with multiple wells. (**A**) Apply single hole puncher to pre-cut parafilm strips to make required number of wells. (**B**) Attach parafilm strips to the Superfrost microscopic slides by pressing and then remove the upper layer. (**C**) Heat slides on the hot plate at medium power, 30 s to 1 min, and remove them from the heater as soon as parafilm melts. (**D**) Liquid evaporation in long term cultures may be prevented by placing the slides into Petri dishes with 10–15 mL DPBS or any similar buffer. (**E**) The levels of cell media in the wells after 24 h of incubation were well maintained. (**F**) After removing parafilm, the surface of slides around the well with cells, are still hydrophobic (fixation solution remains within hydrophilic areas with cells).

**Figure 2 cells-14-01271-f002:**
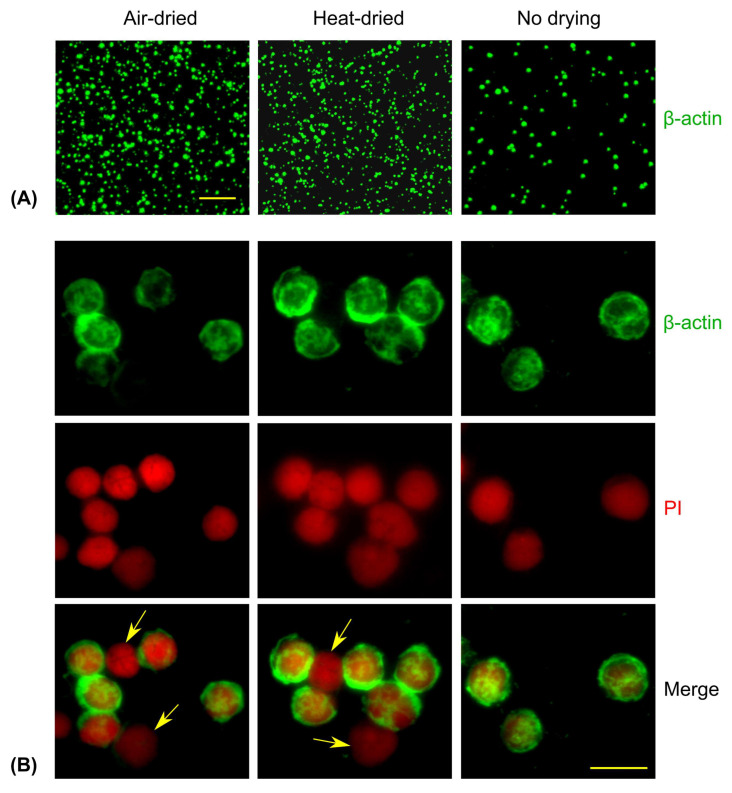
Comparison of air-dried, heat-dried, and immediately fixed slides. Murine splenocytes, incubated for 30 min on microscope slides, were either air-dried (left), heat-dried (middle), or immediately fixed (right). (**A**) Low resolution, scale bar is 200 μm. (**B**) High resolution, scale bar is 10 μm. Green: β-actin, red: PI. Dead cells are positive for PI but negative for β-actin (arrows).

**Figure 3 cells-14-01271-f003:**
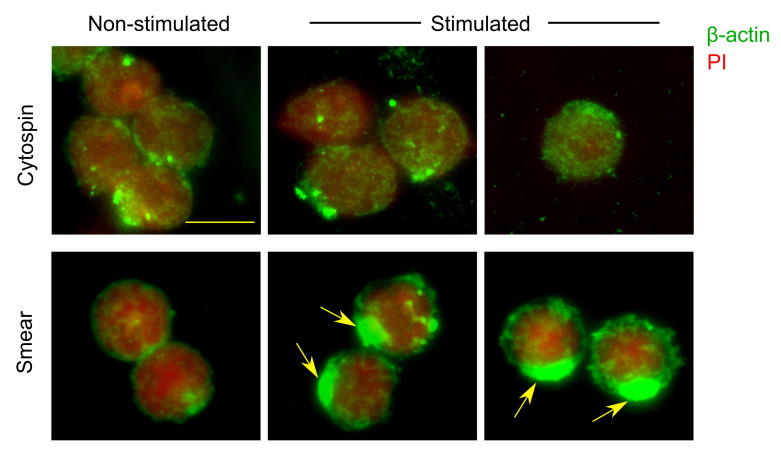
Comparison of cytospin preparation and cell smears: cytoskeletal reorganization. Murine splenocytes were incubated for 3 h, in the presence or absence of PMA and ionomycin. After incubation, cells were split into two portions and prepared via standard cytospin (**top**) or via cell smears (**bottom**), as described in Methods. Arrows: accumulation of β-actin staining at one pole of the cell, visible at smears but not in cytospin preparations. Scale bar is 10 μm. Green: β-actin, red: PI.

**Figure 4 cells-14-01271-f004:**
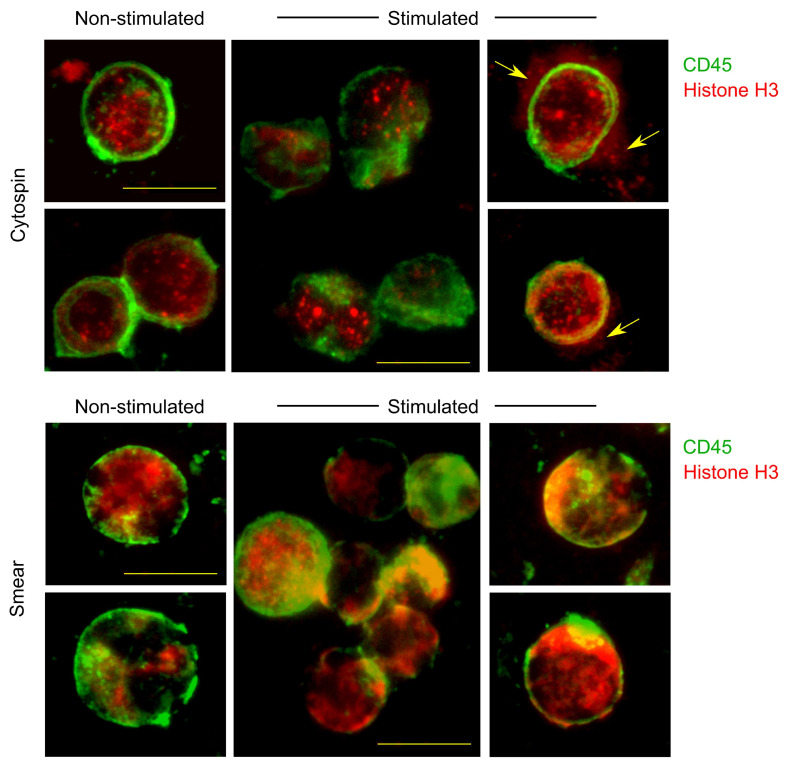
Comparison of cytospin preparation and cell smears: staining artifacts. Murine splenocytes were incubated for 3 h, in the presence or absence of PMA and ionomycin. After incubation, cells were split into two portions and prepared via standard cytospin (**top**) or via cell smears (**bottom**), as described in Methods. Arrows: “leakage” of the histone H3 signal beyond the cellular boundaries, observed in cytospin, but not smear preparations. Scale bar is 10 μm. Green: CD45, red: Histone H3.

**Figure 5 cells-14-01271-f005:**
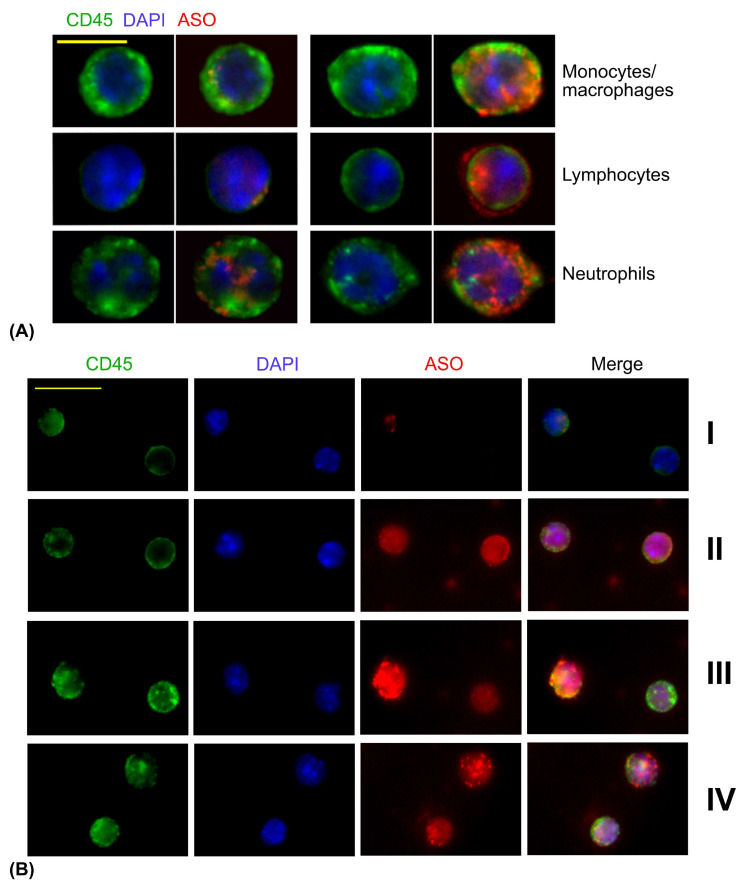
Murine splenocytes and ASO uptake. (**A**) Types of cells that may be identified by morphology: monocytes/macrophages (top), lymphocytes (middle) and neutrophils (bottom). Scale bar is 12.5 μm. (**B**) Different patterns of ASO distribution in primary cells. Row I: few spots of ASO (left) and no ASO uptake on the right. Row II: diffuse ASO signal in both cells, with slightly higher intensity on the right. Row III: predominantly diffuse ASO with high intensity (left) and diffuse ASO with low intensity on the right. Row IV: diffuse ASO with a lot of granules of ASO (top), diffuse ASO with a few granules on the bottom. Scale bar is 25 μm. Green: CD45, blue: DAPI, red: ASO.

**Figure 6 cells-14-01271-f006:**
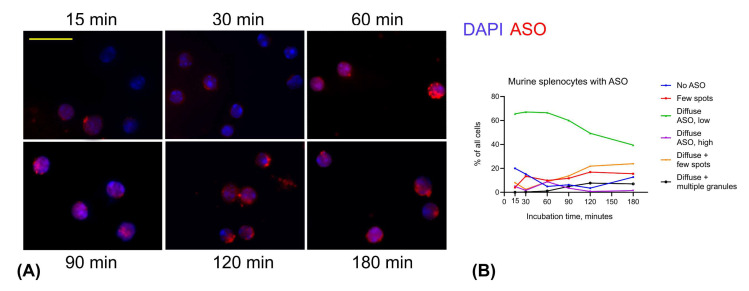
Distribution of ASO in splenocytes and cellular uptake over time. (**A**) Illustrative examples of lymphocytes appearance at different time points. Scale bar is 25 μm. (**B**) Time series graph showing percent of the splenocytes with corresponding pattern of ASO. Blue: DAPI, red: ASO.

**Figure 7 cells-14-01271-f007:**
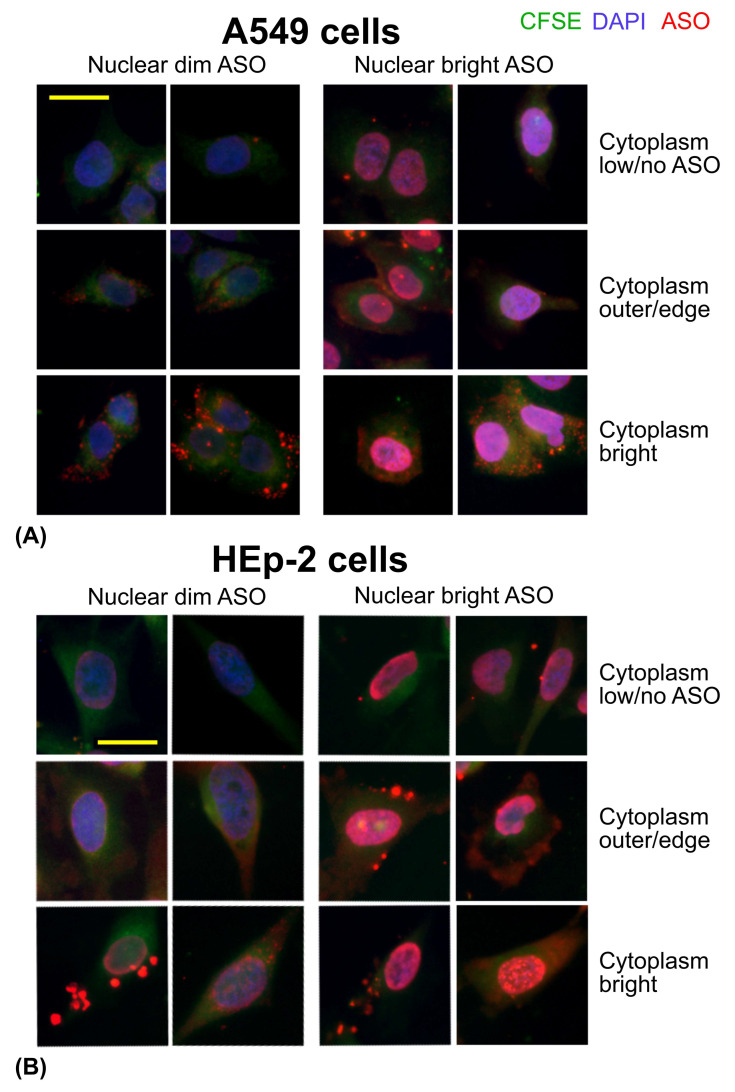
Distribution of ASO in A549 and HEp-2 cells. (**A**) A549 cells. Scale bar is 25 μm. (**B**) HEp-2 cells. Scale bar is 25 μm. Green: CFSE, blue: DAPI, red: ASO.

**Figure 8 cells-14-01271-f008:**
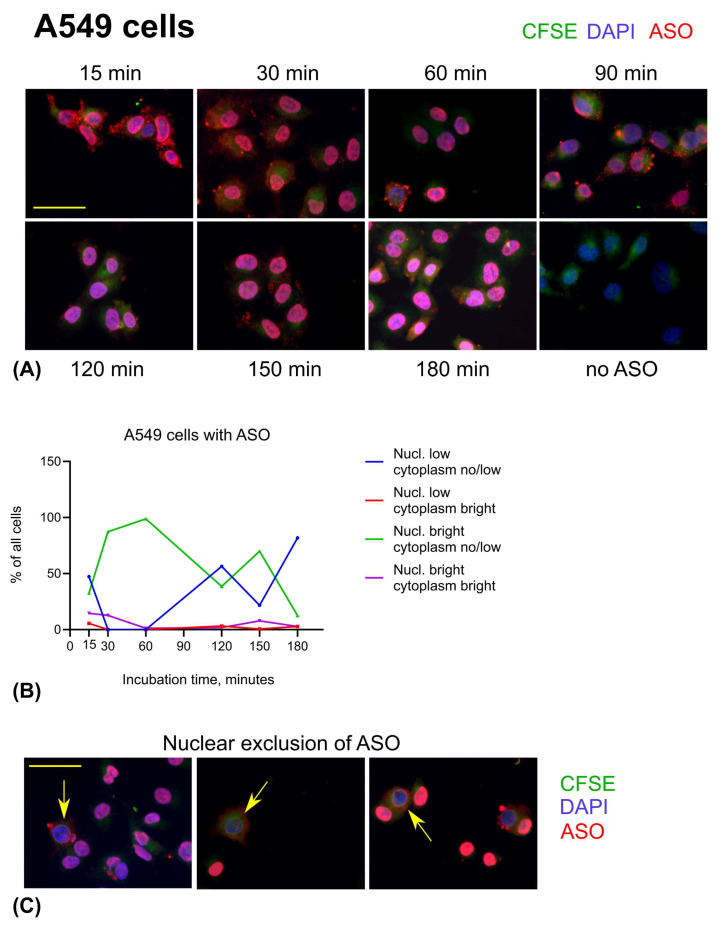
Distribution of ASO in A549 cells over time. (**A**) Examples of A549 cells staining at different time points. Scale bar is 50 μm. (**B**) Time course graph showing percent of the A549 cells with corresponding pattern of ASO. (**C**) Nuclear exclusion of ASO in A549 cells (arrowed). Scale bar is 50 μm. Green: CFSE, blue: DAPI, red: ASO.

**Figure 9 cells-14-01271-f009:**
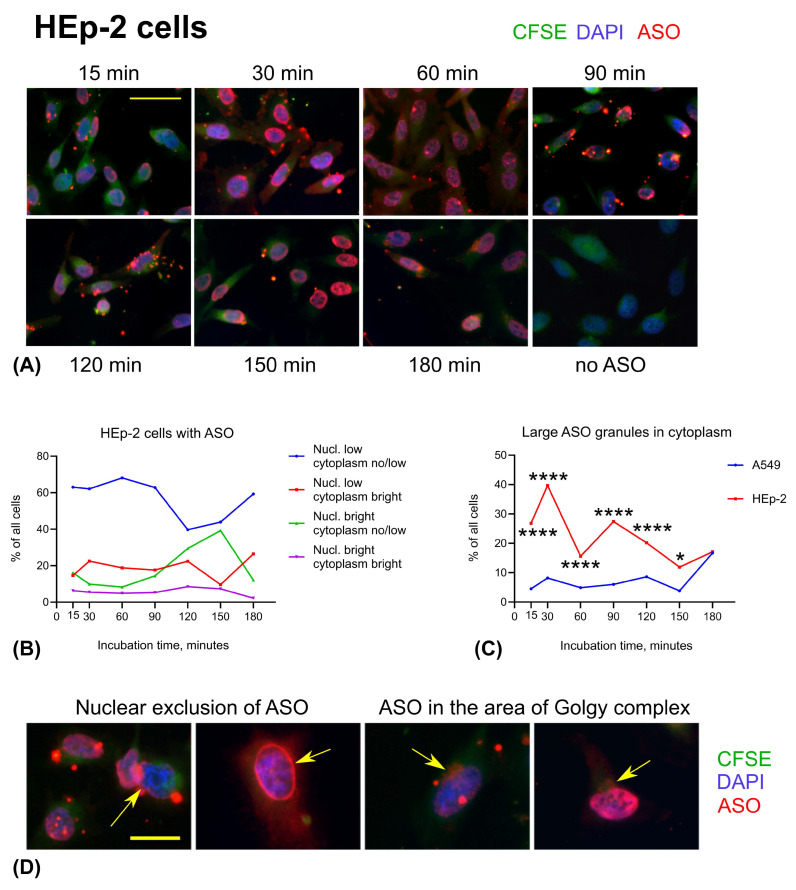
Distribution of ASO in HEp-2 cells over time. (**A**) Illustrative examples of HEp-2 cells appearance at different time points. Scale bar is 50 μm. (**B**) Time series graph showing percent of the HEp-2 cells with corresponding pattern of ASO. (**C**) Occurrence of large cytoplasmic granules with ASO in A549 vs. HEp-2 cells. Cell counts compared by Fisher’s tests separately for each time point, then *p* values were adjusted using Bonferroni correction. **** *p* < 0.0001; * *p* < 0.05. (**D**) Nuclear exclusion of ASO in HEp-2 cells (arrowed, left). Accumulation of ASO in the perinuclear areas corresponding with the location of Golgi complex (arrowed, right). Scale bar is 25 μm. Green: CFSE, blue: DAPI, red: ASO.

## Data Availability

The raw data supporting the conclusions of this article will be made available by the authors, without undue reservation.

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
