# Peer review of "New Approaches to Old Techniques in Cell Handling for Microscopy"

_cells, 2025, doi:10.3390/cells14161271_

Round 1

Reviewer 1 Report

Comments and Suggestions for Authors

This manuscript describes a simplified homemade approach for cell imaging without using a centrifuge or a special chamber slide. The method aims to eliminate the need for selecting appropriate centrifugation time and speed, allowing for compound testing in time series without requiring cytospin or time-lapse imaging systems. While the method appears simple and accessible, there are major concerns that need to be addressed to support the claimed improvements:

  1. There is a heat step used to concentrate cells on the slide. It is unclear how this heat treatment would affect the cells. Although no centrifuge is used, how the heat exposure may stress the cells or interfere with cellular behavior and gene expression is not discussed. This may bias the outcome of the cell-based analyses. Also, the manuscript does not provide clear information about whether this heating step may dry the cells completely. The time and conditions may vary under different environments, depending on the local moisture and airflow. Additional explanation is needed to confirm the reproducibility and safety of this step.
  2. Are the observed cell morphologies comparable to those seen with traditional methods? Without a direct comparison, it is unclear whether this approach affects cell appearance, distribution, or image quality.
  3. It is not clear how the pre-prepared slides were protected from contamination during storage. What precautions were taken? Please specify the storage conditions and how long the slides were stored before use.
  1. It is unclear how cells can be imaged directly on a standard glass slide instead of on a coverslip, which is commonly used for high-quality microscopy. Slides do not provide the same optical compatibility with standard objective lenses. Please explain how this setup affects image resolution and whether this approach supports high-magnification imaging.
  1. The method did not describe how the media or buffer was changed in the wells. How did the authors handle media exchange while keeping the cells in place? Were the cells attached to the slide surface? What measures were taken to prevent cells from being washed away?
  2. What are the differences between Nuclei Bright and Dime-ASO? Were they delivered the same way? What might explain the observed differences between the two? Please clarify the rationale and mechanisms behind the results.
  3. What do the error bars in Figure 5B and Figure 6B represent? Why did the percentage of ASO-positive cells increase at some point and decrease at others? The variability needs to be explained.
  4. What compounds were included in the “ASO uptake” DMEM media? How was the ASO delivered to the cells? Antisense oligonucleotides are typically unable to penetrate cells without a delivery agent or chemical modifications. What was the ASO used in this study? Was it for labeling the membrane or for cellular uptake? Did the ASO enter the cells or simply bind to the cell surface? It would not be a surprise to see that the ASO cannot enter the cell nucleus, and their random attachment to cells was reversible if no delivery reagent were used.
  5. In the cell culture section, “NH4Cl 3 mM and Arsenic 1 μM” should be revised to “3 mM NH4Cl and 1 μM Arsenic.”
  6. Several hyphens (“-”) are used inside words. Please correct these typographical errors throughout the manuscript.

Author Response

We appreciate the time and effort the editors and reviewers have dedicated to evaluating our manuscript. Your detailed and insightful feedback has significantly improved our paper. We have incorporated all suggested revisions. For your convenience, all changes to the manuscript have been marked in red text.  We performed all additional experiments requested by Reviewers, and created new Figures with corresponding text, and made a flow chart with conventional methods along with our new techniques. Below, we provide a point-by-point response addressing each comment and concern raised during the review process.

 Reviewer 1

This manuscript describes a simplified homemade approach for cell imaging without using a centrifuge or a special chamber slide. The method aims to eliminate the need for selecting appropriate centrifugation time and speed, allowing for compound testing in time series without requiring cytospin or time-lapse imaging systems. While the method appears simple and accessible, there are major concerns that need to be addressed to support the claimed improvements:

Comment 1: There is a heat step used to concentrate cells on the slide. It is unclear how this heat treatment would affect the cells. Although no centrifuge is used, how the heat exposure may stress the cells or interfere with cellular behavior and gene expression is not discussed. This may bias the outcome of the cell-based analyses. Also, the manuscript does not provide clear information about whether this heating step may dry the cells completely. The time and conditions may vary under different environments, depending on the local moisture and airflow. Additional explanation is needed to confirm the reproducibility and safety of this step.

Response 1:

The heating step was not intended to concentrate cells on the slide, but rather to fix them. Heat fixation is a well-established, traditional technique used primarily to preserve cellular morphology for microscopy. Due to its long-standing use in laboratory practice, formal references in peer-reviewed publications are limited; the method is more commonly described in student textbooks and widely circulated laboratory protocols. The following textbook may serve as a suitable reference:

Wendy Keenleyside (2019) “2.4 Staining Microscopic Specimens” in “Introduction to microbiology: Canadian Edition", Canadian, Keenleyside, Pressbooks

https://ecampusontario.pressbooks.pub/microbio/,

text: “Fixation is often achieved either by heating (heat fixing) or chemically treating the specimen. In addition to attaching the specimen to the slide, fixation also kills microorganisms in the specimen, stopping their movement and metabolism while preserving the integrity of their cellular components for observation”.

Here: https://www.thermofisher.com/us/en/home/life-science/protein-biology/protein-biology-learning-center/protein-biology-resource-library/pierce-protein-methods/fixation-strategies-formulations.html

“Common methods of fixation include: (…) Drying: Blood smears for ICC staining are air-dried and waved across a flame to heat-fix the cells to the slide”.

Or here:

https://nerd.wwnorton.com/ebooks/epub/microbio6/EPUB/content/2.4-chapter02.xhtml

“A typical procedure for fixation and staining is shown in Figure 2.22. First we fix a drop of culture on a slide by treating it with methanol or by heating it on a slide warmer (steps 1–4 in the figure). Either of these treatments denatures cell proteins, exposing side chains that bind to the glass.”

In other words, although we propose methods suitable for time-lapse series analysis, our intention is not to maintain cell viability. Rather, our objective is to kill, dry, and fix the cells while preserving their morphology. Heat fixation serves this purpose by denaturing cellular proteins and causing them to adhere to the slide surface, thereby preventing their loss during staining procedures. This process also rapidly kills the cells and inactivates proteolytic enzymes, preventing autolysis.

Because our goal is to ensure complete cell death, firm attachment, and thorough drying, the heating step was deliberately optimized to eliminate the influence of local humidity and airflow under typical laboratory conditions. We recommend using mild heat, 55–60°C for 20 minutes. To ensure reproducibility, users should employ a plate heater with temperature control or monitor the surface temperature with a thermometer to confirm it falls within the specified range.

In this study, we validated the reproducibility of the heating procedure across more than 50 slides, using two different cell lines and one set of primary cells. The method was tested in six independent experiments: two with murine splenocytes, two with A549 cells, and two with HEp-2 cells, in addition to several preliminary trials that did not yield optimal results.

Although we did not observe any morphological differences or staining artifacts between heat-dried and air-dried cells, only the dried slides, both air- and heat-dried, retained apoptotic cells. These apoptotic cells were readily identifiable by the absence of β-actin staining in the cytoplasm and the presence of propidium iodide-positive nuclei. This finding may be significant for experiments where the detection or quantification of apoptotic or dead cells is necessary.

We have included this observation in the revised manuscript (new lines 239–252) and added a new figure (Figure 2) to illustrate these results.

Comment 2: Are the observed cell morphologies comparable to those seen with traditional methods? Without a direct comparison, it is unclear whether this approach affects cell appearance, distribution, or image quality.

Response 2: We believe that the quality of the images and the preservation of cellular morphology were clearly demonstrated through the numerous micrographs of individual cells included in our manuscript.

However, in response to this comment, we revisited our earlier protocols, mentioned in the Introduction, that had shown poor cytospin performance, and conducted an additional set of experiments.

Murine splenocytes were incubated in cell culture tubes for 3 hours in the presence or absence of strong stimulation using phorbol 12-myristate 13-acetate (PMA) and ionomycin. Following incubation, the cells were divided into two groups: one group was used to prepare cytospin slides using a standard protocol, while the other group was processed into smears using our current (cytospin-free) method. We then applied two different staining protocols to evaluate and compare cellular morphology and staining quality between the two slide preparation techniques—with and without centrifugal force.

Staining for beta-actin and propidium iodide revealed that both methods preserved general cellular morphology. However, the staining patterns differed slightly between the two approaches. In smear preparations, most stimulated lymphocytes showed a distinct accumulation of bright β-actin staining at one pole of the cells—a pattern not observed in cytospin preparations, likely due to mechanical distortion caused by centrifugal force.

Staining for histone H3 and CD45 demonstrated even more pronounced differences. In cytospin preparations, we observed staining artifacts, including “leakage” of the histone H3 signal beyond cellular boundaries in stimulated splenocytes. This finding aligns with our previous observations, referenced in the Introduction, that the membranes of activated lymphocytes can become fragile and more susceptible to mechanical stress. As a result, cytospin-induced shear forces may contribute to the formation of such artifacts. We reported results of this experiment in the new Figures 3-4, and in the revised manuscript text (lines 253–289).

Comment 3: It is not clear how the pre-prepared slides were protected from contamination during storage. What precautions were taken? Please specify the storage conditions and how long the slides were stored before use.

Response 3: Slides designated for primary cell incubation using a hydrophobic barrier pen should be prepared immediately prior to use, not stored.

The storage of slides with parafilm wells was addressed in the lines 181-183: “A large lot of slides may be prepared in advance, and then stored in closed plastic bags for future use; in that case, rinsing and ultraviolet treatment should be performed prior to cell culture”.

Dry glass slides with attached parafilm, when stored in sealed plastic bags, are not particularly prone to contamination. According to the manufacturer, the shelf life of Superfrost Plus adhesion slides is one year. Therefore, the storage period for the slides with attached parafilm should not exceed the expiration date of the slides, unless their adhesive properties have been experimentally verified beyond that time. In our laboratory, we do not have data on storage beyond one year, as this final version of the parafilm-well setup was developed relatively recently.

Regarding the additional precautions, we tested immersing slides in 70% alcohol and also in 3% bleach solutions, followed by rinsing with DI water. Parafilm wells remained structurally intact, as parafilm is resistant to these solutions, and cells cultured on such slides exhibited no noticeable differences in growth compared to cells on slides without additional sterilization.

In response to this comment, we have added the following clarification to the Methods section (lines 183–186): “When needed, slides with parafilm wells may be washed by immersing them in 70% ethanol or 3% bleach solution, followed by rinsing with deionized water. These procedures do not affect the structural integrity of the wells or the adhesive properties of Superfrost Plus slides.”

Comment 4: It is unclear how cells can be imaged directly on a standard glass slide instead of on a coverslip, which is commonly used for high-quality microscopy. Slides do not provide the same optical compatibility with standard objective lenses. Please explain how this setup affects image resolution and whether this approach supports high-magnification imaging.

Response 4: From a microscopy perspective, there is essentially no difference between growing cells on coverslips versus directly on microscope slides. In both cases, the final configuration is the same: the sample is mounted on a microscope slide, covered with a layer of cells in mounting medium, and topped with a coverslip. Thus, the cells are always positioned between the microscope slide and the coverslip. Standard objective lenses and conventional microscopes are used, with the typical optical path being: microscope slide-> cells -> coverslip -> air (or immersion oil, when applicable) -> objective lens.

Similarly, for tissue sections—whether paraffin-embedded or cryosections—the tissue is affixed to microscope slides, not coverslips. This setup does not pose any issue for high-magnification imaging, as observation is always conducted through the coverslip.

Our parafilm-well system follows the same principle as commercially available chamber slide systems. For reference, please see the diagram under the section “The Principle of the Chamber Slides, removable” here

https://ibidi.com/chamber-slides/44-3-well-chamber-removable.html

Comment 5: The method did not describe how the media or buffer was changed in the wells. How did the authors handle media exchange while keeping the cells in place? Were the cells attached to the slide surface? What measures were taken to prevent cells from being washed away?

Response 5: Adhesive cells attach themselves to the slides, as they usually do. The culture medium or buffer can be easily removed either by gentle shaking, pipetting, or using a vacuum aspirator in the hood. Primary cells, including murine splenocytes and human PBMCs, also adhered surprisingly well to Superfrost Plus slides, although not as firmly as adherent cell lines. This observation is described in the original lines 125–128 and now updated to lines 131–134: “Surprisingly, not only neutrophils, but also lymphocytes, incubated on Superfrost Plus slides for 30 minutes or longer, adhered strongly enough to remain in place when the culture media was gently removed by pipetting from the corner of the slide. Following this step, the slides were heat-fixed and stained as needed.”

No additional special measures were required, gentle pipetting from the corner of the chamber, well, or slide was sufficient, as described in the Methods. After the heating and drying steps, all cell types were firmly attached to the slide surface. However, we did not perform excessive or forceful rinsing during our staining procedures.

Comment 6: What are the differences between Nuclei Bright and Dime-ASO? Were they delivered the same way? What might explain the observed differences between the two? Please clarify the rationale and mechanisms behind the results.

Response 6: As described in Methods section, the ASO was simply added directly to the cell culture medium containing the cells, and the incubation was stopped at the corresponding time points indicated in the figures (new lines 98–103 for primary cells; new lines 210–216 for cell lines). No special delivery method was required for this type of ASO.

The bright and dim nuclear and cytoplasmic ASO staining patterns were distinguished based on relative signal intensity, or by the absence of a bright signal. These differences are clearly illustrated in the new Figure 7 (previously Figure 4): dim nuclei appear almost blue, while bright nuclei display a clearly visible pink-purple hue. Cytoplasmic ASO signal varies as well—from dark green (top rows), to dark green with red staining at the cell periphery (middle rows), to bright red dots or diffuse staining across most of the cytoplasm (bottom rows).

While this classification method is quick and practical for evaluating large numbers of cells, it is inherently somewhat subjective, which is typical for histological assessments. The variation in ASO uptake and intracellular distribution patterns, both among individual cells within the same cell line and between different cell types, likely reflects underlying biological differences. These may include variability in plasma membrane composition, endosomal pathways, chaperone proteins, and receptor expression. However, the precise mechanisms remain poorly understood and are beyond the scope of this manuscript.

We wrote an additional text regarding this matter into Discussion.

Comment 7: What do the error bars in Figure 5B and Figure 6B represent? Why did the percentage of ASO-positive cells increase at some point and decrease at others? The variability needs to be explained.

Response 7: There are no error bars in Figures 5B and 6B (new Figures 8B and 9B), since the series graphs show percent of the cells with corresponding patterns of ASO, as we described in Methods, lines 177-180 (new lines 229-233), and also described in Figure Legends.

Regarding ASO-positive cells, it's important to note that only primary cells displayed a (low) percentage of ASO-negative cells across various time points, whereas both cell lines consistently showed ASO positivity. We've detailed these observations in both the previous manuscript version (old lines 205-207, 236-238) and the current one (new lines 329-332, 359-361).

As previously stated, the individual variability in ASO uptake among different primary cell types, within different cell lines, or even within individual cells of the same cell line remains unexplained. The underlying mechanisms are largely unknown. We wrote an additional text regarding this matter into Discussion.

Comment 8: What compounds were included in the “ASO uptake” DMEM media?

Response 8: We described all compounds in the Methods: “For ASO uptake experiments, we used DMEM (cat# 10565042, Gibco), supplemented with 3% of FBS, with an addition of NH4Cl 3 mM and Arsenic 1 μM to enhance gymnosis, as reported [5]. “, old lines 77-79, new lines 83-86

Comment 9: How was the ASO delivered to the cells?

Response 9: This type of ASO does not require any special delivery, but self-delivered by gymnosis, as we described in the lines 20-22, “We present methods and illustrative examples involving the cellular uptake of self-delivering oligonucleotides in murine splenocytes and in two adherent hu-man tumor cell lines”, and in the old lines 302-303 (new lines 476-477) “We illustrate the use of our methods by detailed evaluation of time-series experiments with cellular uptake of self-delivering AUMsilence ASO”. To clarify this issue, we add the additional text and references, lines 461-470

Comment 10: Antisense oligonucleotides are typically unable to penetrate cells without a delivery agent or chemical modifications. What was the ASO used in this study?

Response 10: Yes, indeed. Cellular uptake remains a significant challenge for common ASOs, especially in non-dividing or resting cells. However, in this study, we utilized a novel ASO type that appears to overcome these uptake issues, both in vivo and in vitro, even in resting primary cells like CD4+FOXP3+ regulatory T cells (Tregs). We have previously reported on the use of these ASOs with murine and human cells (PMID: 39234236). While that publication was not directly relevant to the current manuscript, it includes references to six other published papers detailing the type of ASO used and its cellular uptake. This ASO was previously referred to as “FANA ASO”.

We have identified the ASO in the current manuscript (old line 92, new line 98) as “Far-red fluorescent labeled Scramble ASO (AUMsilence, AUM Biotech, LLC)".

To clarify this question, we added an extra data with references to the new version of manuscript, lines 458-461.

Comment 11 (9): Was it for labeling the membrane or for cellular uptake? Did the ASO enter the cells or simply bind to the cell surface? It would not be a surprise to see that the ASO cannot enter the cell nucleus, and their random attachment to cells was reversible if no delivery reagent were used.

Response 11: ASOs were clearly observed within both the nuclei and cytoplasm of cells, as evident from the provided images. The cellular uptake of these ASOs is consistent with findings from previously published work (referenced above and now incorporated into the Discussion), where uptake was demonstrated by the successful downregulation of target gene expression. For the purposes of the current study, we employed a Scramble control ASO (which does not target gene expression) to enable direct observation of cellular uptake, isolating this process from any effects related to modified gene expression.

Comment 12 (10) In the cell culture section, “NH4Cl 3 mM and Arsenic 1 μM” should be revised to “3 mM NH4Cl and 1 μM Arsenic.”

Response 12: Done

Comment 13 (11) Several hyphens (“-”) are used inside words. Please correct these typographical errors throughout the manuscript.

Response 13 Done, thank you.

Reviewer 2 Report

Comments and Suggestions for Authors

The main problem is that the volume is 30 micro L.  This is a severe drawback for those who use fluorescent probes since a much larger volume is required.  For this reason I would not employ this procedure.

There also seems to be great variability in how the cells are prepared.  The problems noted with HeLa cells (260-275) would likely deter those who utilize such cells, and this would probably apply to other cells as well.  It introduces unknowns in the experiment.

Are small well chambers not available commercially?

A flow chart depicting the usual method of conducting these growth experiments, as well as the new technique with a summary of the savings, may help promote the procedure.

Other issues with the manuscript.

the use round cover slips

A lot of  bal-ance in the text.-dashes appear throughout the manuscript.

 we further elaborated it to create-?

few spots in the raw in the middle-?

(low heat, 55-60°C, 20 min,- does this not kill the cells?

drying with ultraviolet lamp, as above-there is no above!

the previously estab-129 lished protocol-this should be referenced-the action of what happens when this parafilm melts was not clearly described.  Does this form a ridge into which liquid can be poured, like the rim of a bowl?  This seems very unlikely to produce similar results each time.

HEp-2 cells originally thought to be derived from an epidermoid carcinoma of the 260 larynx, but later it was established as HeLa cells (cervical cancer) derivative.- this is not a complete sentence.

Comments on the Quality of English Language

See above

Author Response

We appreciate the time and effort the editors and reviewers have dedicated to evaluating our manuscript. Your detailed and insightful feedback has significantly improved our paper. We have incorporated all suggested revisions. For your convenience, all changes to the manuscript have been marked in red text.  We performed all additional experiments requested by Reviewers, created new Figures with corresponding text, and made a flow chart with conventional methods along with our new techniques. Below, we provide a point-by-point response addressing each comment and concern raised during the review process.

Reviewer #2

Comment 1: The main problem is that the volume is 30 micro L.  This is a severe drawback for those who use fluorescent probes since a much larger volume is required.  For this reason I would not employ this procedure.

Response 1: While we agree that a 30 µL well volume might not suit all users, we personally haven't encountered any issues. All probes for live cells may be diluted correspondingly in the cell media, , offering the potential for substantial cost savings on expensive reagents. The adherent cell lines which we tested, had no stress from growing in that volume, and it’s also very easy to change cell media when needed. We are not aware about specific fluorescent probes that require the large volume to use, but in that case, large volume may be simply added into Petri dishes and cover the whole slide. In that modification, slides may be placed just in the bottom of Petri dishes without plastic “coasters” that we used (Figure 1D).

Comment 2: There also seems to be great variability in how the cells are prepared.  The problems noted with HeLa cells (260-275) would likely deter those who utilize such cells, and this would probably apply to other cells as well.  It introduces unknowns in the experiment.

Response 2: It is hard to address this Reviewer’s comment. Cell lines were prepared exactly as described in Methods, without any variability. Variability of ASO intake may be related with multiple factors, mostly not well characterized yet, and described in our responses to another reviewer. We have not noticed and have not described any problems with HeLa cells or rather with HEp-2 cells, HeLa derivative, in the old lines 260-275 (new lines 394-396), or anywhere else in our text. It is difficult to guess why 100% cellular intake of ASO with variable cellular distributions may deter scientists from utilize those or other cells.

Comment 3: Are small well chambers not available commercially?

Response 3: As we discussed in Introduction, lines 49-53, we are aware about commercial well chambers and we even cited the papers describing them, but we have also explained our rationale: “However, this method becomes time consuming and relatively expensive in extensive experiments when new compounds are tested in time-series. Our current microscope with cell imagining system requires the use of slides faced down, which excludes the imaging of multi-well microplates”. Another words, commercial well chambers are relatively expensive while out method is not. Additionally, as we described, we (as any other users of microscopes requiring the use of slides being faced down) cannot look at the wells in commercial well chambers without disassembling them, even if they have a thin coverglass bottoms. Those factors led us to develop a simple, but convenient method for home made analogues of small well chambers.

Comment 4: A flow chart depicting the usual method of conducting these growth experiments, as well as the new technique with a summary of the savings, may help promote the procedure.

Response 4: We have created a flow chart, depicting the conventional approaches along with our methods (this is a graphical abstract).

Regarding the promotion, we are not interested in promoting the procedure, but rather in reporting our modification, this is a simple DIY project. Because of that, it seems not much appropriate for us to provide a summary of saving in the scientific paper. To address this commentary, below we have calculated estimations of savings for Reviewer:

16 slides with 8 wells cost $341.65

https://www.thermofisher.com/order/catalog/product/154941PK

Which is $2.67 per well

72 slides of Superfrost® Plus cost $103.28

https://www.avantorsciences.com/us/en/product/4646061/vwr-premium-superfrost-plus-microscope-slides

parafilm costs $48 for 38 meters

https://www.sigmaaldrich.com/US/en/product/sigma/hs234526b

72 slides x6 wells results with 432 wells, which is $0.35 per well, which is 7.6 times cheaper per well.

We can also calculate a total for an experiment. An average experiment, as we have presented in the paper, includes 8 conditions/slides, and it was repeated twice = 16 slides (may be less, but in our case we have to stop incubation, so the whole slide served as six biological replicates for the same condition). For commercial well chambers system price would be about $683.33. For our method a price would be $33.60.

Other issues with the manuscript.

Comment 5: the use round cover slips

Response 5: corrected, thank you

Comment 6: A lot of  bal-ance in the text.-dashes appear throughout the manuscript.

 we further elaborated it to create-?

few spots in the raw in the middle-?

Response 6: corrected, thank you

Comment 7: (low heat, 55-60°C, 20 min,- does this not kill the cells?

Response 7: Yes, it has to kill the cells and to dry them. We evaluate dead cells, firmly attached to the slides. Here we use heating as the method of fixation.

Comment 8: drying with ultraviolet lamp, as above-there is no above!

Response 8: Old line 121, new line 126: “DI water followed by drying with ultraviolet lamp, as above”.

Old lines 85-86 (new lines 90-91), located above the line 121 (new line 126): “Microscope slides were labeled, rinsed in DI water and dried lying flat in a Class II biological safety cabinet with its ultraviolet lamp on.”

Comment 9: the previously established protocol-this should be referenced-the action of what happens when this parafilm melts was not clearly described.  Does this form a ridge into which liquid can be poured, like the rim of a bowl?  This seems very unlikely to produce similar results each time.

Response 9: The phrase “the previously established protocol” in 2.4 (now it is 2.6) refers to the previously described protocol in 2.3, as we wrote “Using the previously established protocol for short time culture of primary cells directly on Superfrost Plus microscope slides”. To clarify this, we added a reference to corresponding section in the text.

Regarding the parafilm, when it melts, it becomes soft and adheres firmly to the glass. This creates a small well with highly hydrophobic walls, resulting in strong liquid surface tension. Consequently, liquid can be poured into the well, much like into a small bowl. Parafilm offers a superior ability to retain larger liquid volumes compared to a hydrophobic barrier pen. This method consistently yields reproducible results, as the uniform diameter of the hole puncher and consistent material ensure equal well volumes.

Hole punchers are available in various diameters, allowing users to create wells with smaller or larger volumes and diameters as needed:

https://a.co/d/dAQ4Hil

even square wells are possible:

https://a.co/d/5gvq5ry

 In our setup, the actual volume can be increased to 40-50 uL, but we reported using 30 uL as it is safer due to the risk of accidental shaking during manipulations.

We have added information regarding the well diameter (6 mm) in our setup and included additional text on how to further modify this approach if necessary.

Comment 10: HEp-2 cells originally thought to be derived from an epidermoid carcinoma of the 260 larynx, but later it was established as HeLa cells (cervical cancer) derivative.- this is not a complete sentence.

Response 10: Corrected, thank you

Round 2

Reviewer 1 Report

Comments and Suggestions for Authors

The authors did address the majority of my concerns. But I am still concerned about the following two points:

  1. The effectiveness of ASO delivery. Are these cells heat-treated before incubating with ASO? I don't think unmodified ASO can be absorbed by live cells without a delivery reagent. This is demonstrated by many labs, including ours, in delivering DNA and RNA into live cells.
  2. Since the cells are fixed by the heat, the authors should be cautious about claiming time-dependent research with their method. 

Author Response

Thank you for your time and your interest in our work.

Comment 1: The effectiveness of ASO delivery. Are these cells heat-treated before incubating with ASO? I don't think unmodified ASO can be absorbed by live cells without a delivery reagent. This is demonstrated by many labs, including ours, in delivering DNA and RNA into live cells.

Response 1: The cells were not heat-treated prior to incubation with ASO.

As described in the Methods section, the experimental sequence was as follows:

  1. a) Live cells were treated with ASO, either on slides or in cell culture tubes

b1) For cells treated on slides, incubation was stopped at specific time points after start of ASO exposure (e.g., after 15 min, 30 min, 60 min, etc.).  That’s why those experiment are time-dependent or time lapse series, accessing ASO uptake over time. After incubation, cells were fixed via one of three methods: heat, air-drying, or immediate chemical fixation, as described in the experiments corresponding to Figure 2. We found that heat fixation produced the most consistent results but compared all three methods as requested.

b2) For cells treated in cell culture tubes, aliquots were taken at certain time points after beginning of incubation (after 15 min, 30 min, 60 min, etc.), and smears were prepared as described. Then cells were subsequently fixed by heat-drying to prepare them for microscopy. This approach also allowed us to evaluate time-dependent ASO uptake and its intracellular distribution.

Regarding the nature of the ASO: as previously stated in both our initial response to reviewers and in the Discussion, we are not using unmodified ASO. Our ASO is chemically modified and is neither DNA nor RNA. Its ability to enter cells without transfection agents (gymnosis) has been confirmed by several peer-reviewed studies, including our own. These chemical modifications are detailed in the revised Discussion (lines 459–462), and we have cited four relevant references.

For further independent confirmation, we encourage the reviewer to consult members of the Scientific Advisory Board at AUM LifeTech

https://aumlifetech.com/sample-page/contact/scientific-advisory-board/

or to contact the authors of previously published studies involving this type of ASO (commonly referred to as FANA). A list of publications can be found here:

https://www.aumbiotech.com/publications

Alternatively, we invite the reviewer’s group to test the reagent directly. AUMsilence ASOs are commercially available and self-delivery can be readily verified. In our protocol, we included 3 mM NHâ‚„Cl and 1 µM arsenic trioxide to enhance uptake, particularly in primary cells. For cell lines, these additives are often unnecessary; decreasing serum to 3% in the culture medium is typically sufficient.

Comment 2: Since the cells are fixed by the heat, the authors should be cautious about claiming time-dependent research with their method.

Response 2:

We agree that time-resolved studies involving fixed cells require careful interpretation. However, our experiments were designed with this consideration in mind.

Cells were incubated while alive with ASO, and fixation (by heat, air-drying, or chemical means) occurred only after defined incubation periods. This approach is a standard strategy for performing time-dependent endpoint analyses, where cells are fixed at various time points to track dynamic processes.

We also acknowledge that a live-cell imaging system would provide an alternative method for studying ASO uptake in real time. This would require chamber slides with a coverslip bottom compatible with our microscope's inverted optics (slides have to be faced down in our microscope). While we have not yet implemented this setup, it is technically feasible and may be explored in future studies.

Reviewer 2 Report

Comments and Suggestions for Authors

Comments were suitably addressed.

Author Response

Thank you for your time and for your interest in our work.